# A Structurally Characterized *Staphylococcus aureus* Evolutionary Escape Route from Treatment with the Antibiotic Linezolid

Laura Perlaza-Jiménez,[a,b] Kher-Shing Tan,[a,b] Sarah J. Piper,[c,d] Rachel M. Johnson,[c,d] Rebecca S. Bamert,[a,b] Christopher J. Stubenrauch,[a,b] Alexander Wright,[e] David Lupton,[e] Trevor Lithgow,[a,b] Matthew J. Belousoff[b,c,d]

aCentre to Impact AMR, Monash University, Clayton, Victoria, Australia
bInfection Program, Biomedicine Discovery Institute and Department of Microbiology, Monash University, Clayton, Victoria, Australia
cDrug Development Biology, Monash Institute of Pharmaceutical Sciences, Parkville, Victoria, Australia
dCentre for Cryo-electron Microscopy of Membrane Proteins, Monash Institute of Pharmaceutical Sciences, Monash University, Parkville, Victoria, Australia
eSchool of Chemistry, Monash University, Clayton, Victoria, Australia

Laura Perlaza-Jiménez and Kher-Shing Tan contributed equally to the article. The order of names was determined alphabetically.

**ABSTRACT** Methicillin-resistant *Staphylococcus aureus* (MRSA) is a bacterial pathogen that presents great health concerns. Treatment requires the use of last-line antibiotics, such as members of the oxazolidinone family, of which linezolid is the first member to see regular use in the clinic. Here, we report a short time scale selection experiment in which strains of MRSA were subjected to linezolid treatment. Clonal isolates which had evolved a linezolid-resistant phenotype were characterized by whole-genome sequencing. Linezolid-resistant mutants were identified which had accumulated mutations in the ribosomal protein uL3. Multiple clones which had two mutations in uL3 exhibited resistance to linezolid, 2-fold higher than the clinical breakpoint. Ribosomes from this strain were isolated and subjected to single-particle cryo-electron microscopic analysis and compared to the ribosomes from the parent strain. We found that the mutations in uL3 lead to a rearrangement of a loop that makes contact with Helix 90, propagating a structural change over 15 Å away. This distal change swings nucleotide U2504 into the binding site of the antibiotic, causing linezolid resistance.

**IMPORTANCE** Antibiotic resistance poses a critical problem to human health and decreases the utility of these lifesaving drugs. Of particular concern is the "superbug" methicillin-resistant *Staphylococcus aureus* (MRSA), for which treatment of infection requires the use of last-line antibiotics, including linezolid. In this paper, we characterize the atomic rearrangements which the ribosome, the target of linezolid, undergoes during its evolutionary journey toward becoming drug resistant. Using cryo-electron microscopy, we describe a particular molecular mechanism which MRSA uses to become resistant to linezolid.

**KEYWORDS** antimicrobial resistance, MRSA, antibiotics, ribosomes, cryoEM, *Staphylococcus aureus*, antibiotic resistance, drug resistance evolution, electron microscopy, linezolid

Bacterial infections are a massive burden on modern health care, exacting significant human and financial costs (1). Treatment of bacterial infections with antibiotics is becoming increasingly compromised due to the evolution of antimicrobial-resistant (AMR) phenotypes in bacteria such as *Staphylococcus aureus*, which appears in many nosocomial and wound-related infections (2). The Centers for Disease Control rates the evolution of AMR phenotypes in *S. aureus* as an urgent threat to human health, and the rapid evolution of methicillin-resistant *S. aureus* (MRSA) coupled with high rates of community-acquired MRSA promotes the need to use more potent antibiotics. Linezolid was the first fully synthetic antibiotic, introduced for clinical use in 2000 (3, 4). Due to observations of sporadic

Address correspondence to Trevor Lithgow, trevor.lithgow@monash.edu, or Matthew J. Belousoff, matthew.belousoff@monash.edu.

The authors declare no conflict of interest.

outbreaks of linezolid resistance, further compounds of the oxazolidinone class of antibiotics, such as tedizolid, radezolid, and contezolid, are now being developed. However, MRSA clones are evolving resistance to these newer-generation oxazolidinones as well (3, 5, 6), requiring a reevaluation of how, and how readily, *S. aureus* accommodates and breaks through the bacteriostatic effect of oxazolidinone antibiotics. Linezolid and tedizolid resistance are considered to be uncommon relative to other AMR phenotypes in MRSA (5, 7), providing limited opportunities to understand any constraints and the relative fitness of strains resistant to oxazolidinone antibiotics.

Linezolid inhibits ribosome function at the peptidyl transferase center by sterically blocking the 'A-site' for incoming 3′-amino acylated tRNA docking into the 50S subunit of the bacterial ribosome (8–12). Thus, the presence of linezolid directly impacts the core process of protein synthesis. Studies with MRSA *in vitro* have revealed that short-term exposure to linezolid induces a stress response, resulting in an initial adaptation to the drug (13). This adaptive response is characterized by the transcriptional profiles of 18 small RNAs (sRNAs) when MRSA is grown for 30 min in the presence of sub-MICs of linezolid (13). While none of these sRNAs impact the growth rates of MRSA in rich brain heart infusion (BHI) medium, nor do they change the linezolid sensitivity as determined by MIC assays, the expectation is that these sRNAs mediate a network of proteomic changes to protect MRSA during growth in the presence of the antibiotic. Whole-genome sequencing (WGS) studies investigating longer-term exposure to linezolid in the context of human treatments have revealed that the mutations which cause linezolid resistance are found only in genes encoding the 23S rRNA (14). There have also been reports of mutations in the ribosomal proteins uL4 and uL3 (15) that can generate linezolid resistance, reviewed by Long et al. (16). Recent studies carried out in human patients suggest that, in the course of linezolid treatment for a MRSA infection, as little as 14 days is required to see a linezolid-resistant clone evolve and take over the MRSA population at the infection site (14, 17). Across 32 isolates subjected to WGS analysis in one study in Taiwan, linezolid resistance was attributed to a G2576U (*Escherichia coli* numbering of rRNA nucleotides used throughout) variant in the 23S rRNA (14). Additionally, a G2576U variant independently evolved in a human infection study in France (17) and in another study in Japan (18). This nucleotide is located in the peptidyl transferase center and directly impacts the shape of the linezolid-binding site (11). A meta-analysis published in 2020 which assessed global clinical data on MRSA infections suggested that linezolid resistance occurs rarely (19).

While resistance to linezolid may be considered a rare event (5, 7, 19), even a slow spread of linezolid resistance is of concern given its status as a last-line antibiotic. In this study, we used stepwise *in vitro* selection for linezolid resistance in MRSA. WGS analysis revealed mutations corresponding to linezolid resistance in clones which had fitness defects readily detected through growth phenotype assays. To understand the mechanism by which linezolid resistance is imparted, we employed single-particle cryo-electron microscopy (cryoEM) analysis of the 50S ribosomal subunit and showed that the mutations in the gene encoding ribosomal protein uL3 ultimately delivered the same structural defect as reported for mutations in the 23S rRNA suggesting that, in structural terms, a common escape mechanism defines linezolid-resistant MRSA.

## RESULTS

To assess the prominent mutational landscape around linezolid resistance, we performed a selection experiment on *S. aureus* ATCC 43300, a type strain of MRSA that is sensitive to oxazolidinone antibiotics. This strain is in clinical use as a reference standard for antimicrobial susceptibility testing. The selection experiment was performed using a standard brain heart infusion (BHI) broth, where a static culture was inoculated with $10^8$ CFU of MRSA and treated with linezolid at levels below the MIC (Fig. 1A). Once growth was observed in the static culture, selection was applied by streaking onto BHI agar plates containing concentrations of linezolid above or below the MIC. If colonies grew on the selection plates, they were recultured and stored, with at least 3 of these colonies used for subsequent rounds of selection at higher concentrations of linezolid (Fig. 1A). In this way, there were three "generations" of

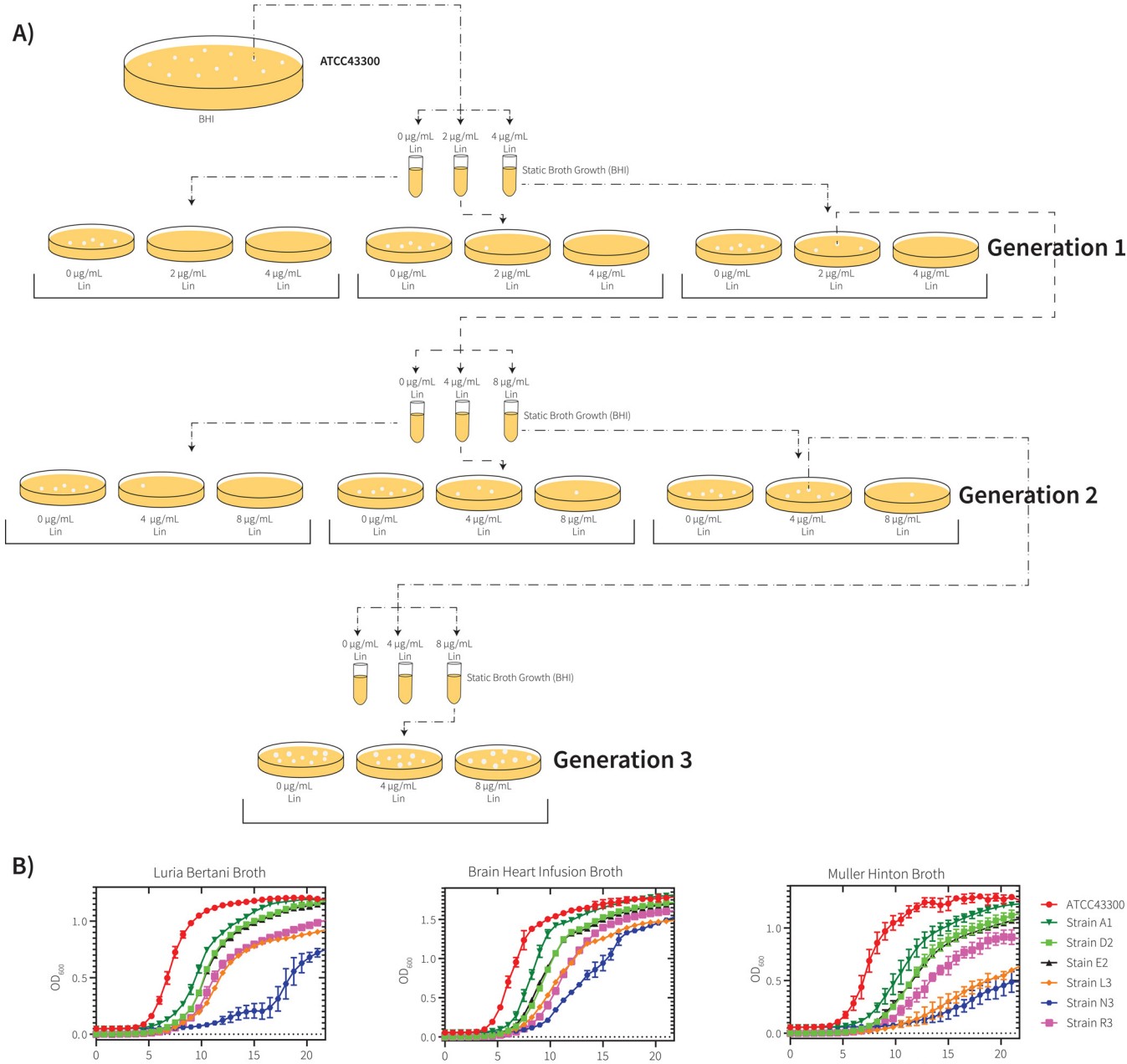

**FIG 1** Selection for linezolid antibiotic resistance in *S. aureus*. (A) Schematic of linezolid resistance selection experiment. A single colony of *S. aureus* (ATCC 43300) was grown under static growth conditions at 37°C until growth was observed. These were then plated on brain heart infusion (BHI) agar plates at increasing linezolid concentrations. If colonies formed on the agar plates, the strain was then passaged through another round of static growth at various linezolid concentrations and the process was repeated until there was confluent growth on 8 $\mu$g/$\mu$L linezolid BHI agar. Clones were subject to genomic DNA (gDNA) sequencing at each of the three generations that made up the experiment. (B) Growth curves of the ancestral strain ATCC 43300 and the evolved strains in Luria Bertani broth, BHI broth, and Mueller-Hinton broth.

strains, corresponding to the three drug-resistance selection steps. For the naming convention of the selected clones, we used A1/B1/C1 for Generation 1, D2/E2/F2/G2/H2/J2 for Generation 2, and K3/L3/N3/M3/O3/P3/R3 for Generation 3 strains (see Fig S1 in Text S1 in the supplemental material).

Clones of MRSA isolated through drug selection were tested for growth in three relevant bacterial media: BHI broth, Luria-Bertani (LB) broth, and Mueller-Hinton (MH) broth. After selection on linezolid, there was a significant delay in growth on BHI medium for the Generation 1 strain A1 and a small but significant defect in growth rate. The growth defect observed for strain A1 was lessened in LB and MH media (see Data Set S2 in the supplemental

material), consistent with a fitness cost that is partly ameliorated in rich growth medium. The Generation 2 strains had a further drop in $k_{obs}$ values (Data Set S2) on all three growth media, despite showing a lag phase of a similar duration to that of strain A1 (Fig. 1B). For the Generation 3 strains, L3, N3, and R3 represent the three different lineages (see figure in Text S1 in the supplemental material), and in all three cases showed growth defects which were even more profound in all three growth media (Fig. 1B). Strain N3 exhibited the slowest $k_{obs}$ and the longest lag time in comparison to strains L3 and R3. These data extend and quantify previous observations of slow growth for linezolid-resistant MRSA strains which have a small colony phenotype when plated on solid media (14, 17).

Seventeen of the evolved clones (three from the first generation, six from the second, and eight from the third) were subjected to WGS using Illumina short-read sequencing. A number of single-nucleotide variations (SNVs) were identified in each genome (see Data Set S2 in the supplemental material). To be certain that these had arisen in the course of this experiment, we sequenced the ancestral clone of ATCC 43300 used in this study in parallel for comparison. A total of 243 SNVs were identified in the alignment of all strains using ATCC 43300 as a reference. There were no physical hot spots for mutation, with the SNVs distributed across the genome (Fig. 2A).

From all SNVs observed, 135 changes were nonsynonymous, resulting in changes to protein sequences. The total number of SNVs per strain, including those that occur in intergenic regions, varies between 57 and 96 (Table 1). The appearance of SNVs in different locations in the genome seemed to occur at random in each generation, with some SNVs then becoming fixed over time. Most of these fixed SNVs were nonsynonymous changes, consistent with positive selection favoring the protein structural changes caused by these mutations. These fixed SNVs occurred in genes encoding proteins of diverse function: deoxyribose-phosphate aldolase, a prophage terminase small subunit, the oligopeptide transport system component OppA, the acetolactate synthase AlsS, the 50S ribosomal protein uL3, and the acetolactate/zinc metalloproteinase aureolysin. Two of the Generation 3 strains had accumulated the most mutations, with 96 in strain L3 and 94 in strain R3. One striking finding was that, in order to establish themselves for growth in the presence of linezolid, the three Generation 1 strains have a total of 45 SNVs in common. The nonsynonymous changes were inherited and maintained in subsequent generations (Fig. 2A).

Only in one case did we identify two SNVs in a single open reading frame, these occurring in the gene encoding ribosomal protein uL3 (Fig. 2B). The 70S ribosome in *S. aureus* is composed of the 5S, 16S, and 23S rRNA, and 46 ribosomal proteins (11, 20). Analysis of the rRNA sequences and the open reading frame sequences encoding the 46 ribosomal proteins identified only a single mutation in the Generation 1 strains A1, B1, and C1, corresponding to a G155R change in ribosomal protein uL3 (Fig. 2B). This G155R mutation has been reported to confer resistance to the pleuromutilin drug tiamulin in *S. aureus* (21, 22). The clinical breakpoint is defined as the concentration of antibiotic used to define whether an infection is likely to be treatable in a patient, and for MRSA, it is 8 $\mu$g/mL. By Generation 3, strains of MRSA had been evolved with a phenotype that exceeded this clinical breakpoint (MIC of 8 $\mu$g/mL; Table 1). One of these, strain N3, provided a unique opportunity. All of its nonsynonymous SNVs were already present in its Generation 1 ancestor (Fig. 2A, Fig. S1 in Text S1), but N3 had also acquired a second M169I mutation in ribosomal protein uL3 (Fig. 2B). Given the poor growth rates of N3 compared to those of its ancestors E2 and A1 (Fig. 1B), we sought to understand the impact of these mutations on the ribosome.

70S ribosomes were isolated from both the ancestral MRSA strain ATCC 43300 and strain N3. Both samples of ribosomes were subjected to single-particle cryoEM, being imaged in a cryoTEM equipped with a direct electron camera. In the context of 70S ribosome particles, a final masked refinement of the 50S subunits using the RELION software package (23) yielded maps resolved to 3.0 Å for the ancestral ribosome structure and 2.9 Å for the ribosome structure for strain N3 (see Fig. S2 in Text S1 and Table 2). This map resolution is high enough for unambiguous assignments for the mutation site and the nearby peptidyl transferase center (Fig. 3).

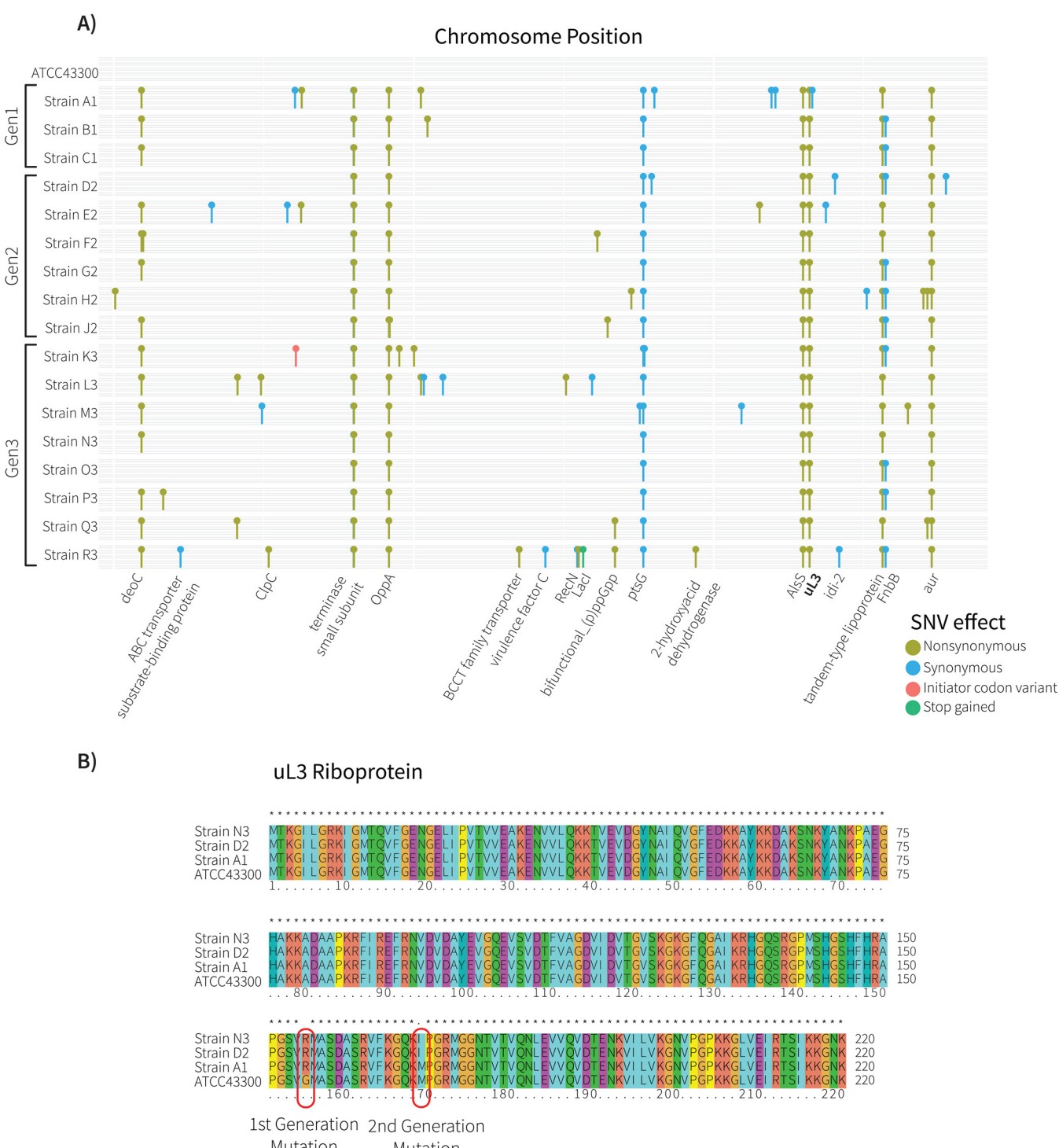

**FIG 2** Mutations that give rise to linezolid-resistant phenotypes. (A) Distribution of single-nucleotide variations (SNVs) in the genome in comparison with ATCC 43300. Each linear row represents the aligned chromosome for each of the 17 strains. Only SNVs which occurred within a protein coding sequence are shown. Each SNV was evaluated for its effect on gene function: nonsynonymous (i.e., changing the encoded protein sequence), synonymous (i.e., not changing the encoded protein sequence), and stop-gained (i.e., nonsense mutation) are indicated by the legend. In one case, an initiator (start) codon was introduced in the gene *AdhP* in strain K3. (B) Multiple sequence alignment of ribosomal protein uL3 sequences from each generation that exhibited linezolid resistance. Two mutations were observed: the first (G155R) appeared in Generation 1, giving rise to moderate linezolid resistance, and the second (M169I) appeared in Generations 2 (under selection with 4 μg/mL linezolid) and 3 (under selection with 8 μg/mL linezolid).

**TABLE 1** SNVs and MIC assessments for the evolved MRSA strains[a]

| Strain | No. of SNVs | MIC (μg/mL) | Clinical designation | uL3 mutation |
|---|---|---|---|---|
| ATCC 43300 | | 1 | Susceptible | None |
| A1 | 89 | 4 | Susceptible | G155R |
| B1 | 72 | 4 | Susceptible | G155R |
| C1 | 71 | 4 | Susceptible | G155R |
| D2 | 76 | 4 | Susceptible | G155R/M169I |
| E2 | 70 | 4 | Susceptible | G155R/M169I |
| F2 | 65 | 4 | Susceptible | G155R/M169I |
| G2 | 72 | 4 | Susceptible | G155R/M169I |
| H2 | 84 | 4 | Susceptible | G155R/M169I |
| J2 | 74 | 4 | Susceptible | G155R/M169I |
| K3 | 89 | 8 | Resistant | G155R/M169I |
| L3 | 96 | 8 | Resistant | G155R/M169I |
| M3 | 62 | 8 | Resistant | G155R/M169I |
| N3 | 57 | 8 | Resistant | G155R/M169I |
| O3 | 66 | 8 | Resistant | G155R/M169I |
| P3 | 73 | 8 | Resistant | G155R/M169I |
| Q3 | 74 | 8 | Resistant | G155R/M169I |
| R3 | 94 | 8 | Resistant | G155R/M169I |

[a]SNV, single-nucleotide variation.

The impact of the G155R/M169I double mutation on the loop of ribosomal protein uL3 which contacts rRNA Helix 90 is striking (Fig. 3A to E). Most notable is the drastic change in the overall fold in the loop around the G155R position (Fig. 3C and D). The impact of this change in ribosomal protein uL3 forces Helix 90 of the 23S rRNA to form such that C2512 is moved out of its helical arrangement (Fig. 3D). The mutation M169I is on the other side of the loop in ribosomal protein uL3. The overall effect of the M169I mutation is a further rearrangement around C2512, propagating the structural change further along Helix 90 and culminating in an adjustment of the position of U2504 (Fig. 3E and F). U2504 plays a vital role in protein synthesis and is a key residue in the oxazolidinone binding site, forming part

**TABLE 2** Data collection and refinement statistics

| Data collection | ATCC 43300 | Strain N3 |
|---|---|---|
| Micrographs | 2,583 | 2,749 |
| Electron dose (e⁻/A²) | 47.5 | 47.5 |
| Voltage (kV) | 200 | 200 |
| Exposure time (s) | 40 | 40 |
| Detector | Falcon3 | Falcon3 |
| Pixel size (Å) | 0.895 | 0.895 |
| Defocus range (μm) | 0.5−1.5 | 0.5−1.5 |
| Symmetry imposed | C1 | C1 |
| Particles (final map) | 157 k | 307.6 k |
| Resolution (0.143 FSC) (Å) | 3.0 | 2.9 |
| | | |
| Refinement | | |
| $CC_{map\_model}$ | 0.80 | 0.78 |
| Map sharpening B factor (Å²) | −54 | −41 |
| | | |
| Model quality | | |
| *R.M.S. deviations* | | |
| Bond length (Å) | 0.002 | 0.002 |
| Bond angles (°) | 0.447 | 0.525 |
| *Ramachandran* | | |
| Favoured (%) | 96.43 | 95.70 |
| Outliers (%) | 0 | 0 |
| | | |
| Rotamer outliers | 0 | 0.04 |
| C-beta deviations (%) | 0 | 0 |
| Clashscore | 6.24 | 3.89 |
| RNA backbone outliers (%) | 14 | 15 |

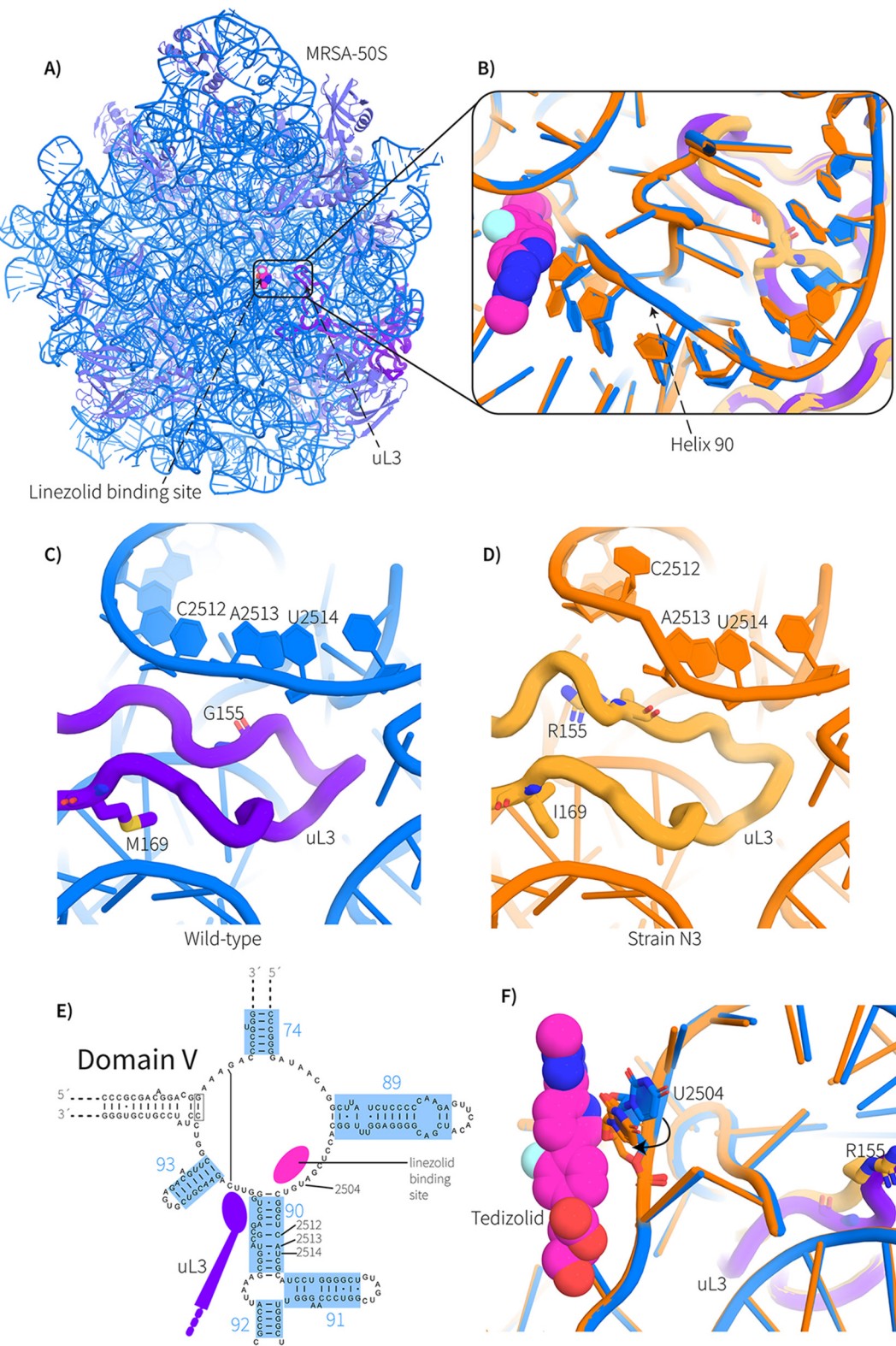

**FIG 3** CryoEM structural analysis of the 50S ribosome of *S. aureus*. (A) Crown view of the 50S ribosome from *S. aureus* ATCC 43300. A molecule of linezolid has been modeled to illustrate the oxazolidinone binding site, as previously defined (50). The rRNA is shown in blue, the ribosomal proteins in magenta, and ribosomal protein uL3 in deep purple. (B) Magnification of the linezolid binding site, with the linezolid model drawn as spheres. The ATCC 4330 (blue, with uL3 shown in purple) and strain N3 structures (orange, with uL3 shown in straw) are overlaid. Note the structural propagation along Helix 90 all the way to the peptidyl transferase center and linezolid binding site. (C) View of the ATCC 43300 ribosome structure centered on the ribosomal

of the A-site wall in the peptidyl transferase center (24). This structural change leaves no space for linezolid to bind the ribosomes in strain N3.

## DISCUSSION

In the original concept for this study, we planned to select for single-step mutants on inhibitory concentrations of linezolid. In this early work, we failed to recover any mutants of ATCC 43300 that could grow on 8 $\mu$g/mL linezolid. We therefore modified the selection protocol to follow multiple steps of less-severe drug selection. One major finding with this modified procedure is a set of common SNVs that appeared independently in the Generation 1 strains A1, B1, and C1, explaining the failed one-step selection. A low level of resistance was conferred by selection on 2 $\mu$g/mL linezolid, and in all cases the strains had acquired 12 common, nonsynonymous SNVs. We were unable to infer the mechanism by which these multiple changes combined to deliver the AMR phenotype. However, the large number of changes required goes some way toward explaining why linezolid resistance was not observed with long-term growth at high (8 $\mu$g/mL) linezolid concentrations; it would be unlikely to see co-selection for so many independent mutations under severe bacteriostatic pressure. This is also a promising observation in terms of treatment options using linezolid on sensitive strains of MRSA.

Ribosomal protein uL3 appears to be a key element for imparting linezolid resistance. All eight of the Generation 3 strains had acquired a M169I mutation in ribosomal protein uL3, in addition to the G155R mutation found in the Generation 1 strains. The G155R mutation at uL3 has been observed previously in linezolid-resistant clones of *S. aureus*, leading to speculation based on their impact on ribosome structure (25, 26), with the data here (Fig. 3) now providing that structural explanation. One of those previous studies reported a linezolid-resistant mutant with a G155R/M169L double mutation in uL3 (25, 26), similar to the G155R/M169I double mutation we detected in strain N3 (Fig. 2). The mutation at position M169 thus has some flexibility regarding which hydrophobic amino acids (L or I) can be tolerated. Distinct uL3 mutations have also been identified and shown to confer linezolid resistance (11): a deletion of residue S145 from ribosomal protein uL3 caused a contraction in the uL3 loop 6, leading to a rearrangement along the rRNA Helix 90, rendering a strikingly similar structural propagation effect on the surrounding rRNA to that observed in our study. In the Generation 2 strains, the G155R/M169L double mutation was present, but the strains did not grow on 8 $\mu$g/mL linezolid. Rather, these strains showed a MIC value of one dilution-step less, i.e., at 4 $\mu$g/mL linezolid, in the MIC dilution series. We have no explanation for this accentuation of the linezolid-resistance phenotype in Generation 3, but it does not correlate to a common genetic change. While we have not tested for any epigenetic effects, we note that hypermethylation of the *rrl* locus can provide increased linezolid tolerance by an unknown mechanism (27, 28).

One limitation of this study is that the strain lineages had no means to acquire new genes via horizontal gene transfer that might accentuate their AMR phenotype. Linezolid-resistant phenotypes have been reported to be enhanced by acquisition of new genes such as *cfr*, which encodes a methylase that modifies A2503 in the 23S rRNA (27, 28). However, our study does provide new information to suggest that an important determinant in the early stages of antibiotic escape is mediated through SNV mutations in ribosomal protein uL3 that can transmit physical changes through the rRNA to rearrange the linezolid binding

**FIG 3** Legend (Continued)

protein uL3 (purple) loop that makes contact with Helix 90 (blue). Note that the G155 makes close contact with A2513 (*E. coli* numbering). (D) The same viewpoint for the strain N3 ribosome structure. The uL3 protein loop is highlighted in straw yellow and the mutation sites are shown by stick representation. Note that the G155R mutation changes the overall topology of the N-terminal side of the uL3 loop, which changes its interaction with Helix 90, introducing a rearrangement of C2512. (E) Two-dimensional map of Domain V of the 23S rRNA, focused around the peptidyl transferase center, indicating the positions of nucleotides C2512, A2513, U2514, and U2504. rRNA helix numbers are in blue and the position of uL3 is in purple. (F) Overlay of the ATCC 43300 (blue/purple) and strain N3 ribosome (orange/straw) structures, centered on the oxazolidinone binding site. A molecule of tedizolid (not present in the structure) has been modeled to illustrate the oxazolidinone binding site, as previously defined (12). Shown is nucleobase U2504, which has moved out to where the oxazolidinone core binds, reducing the steric volume available for oxazolidinone binding.

site. The acquisition of the same SNVs to mutate ribosomal protein uL3 in multiple lineages indicates the penetrance of this change in phenotype. One prior study is consistent with these suggestions, wherein Locke et al. (29) reported mutations in ribosomal protein uL3 in combination with the acquisition of the *cfr* gene in linezolid-resistant *Staphylococci*. Structural analysis suggests that all of these genotype features contribute to linezolid resistance through a rearrangement of the binding sites of linezolid and other oxazolidinones (11, 12, 30). Our study further suggests that this structural rearrangement comes with a concomitant cost to bacterial fitness, impinging upon the function of the peptidyl transferase center and therefore impacting protein synthesis and bacterial growth rate. New combination therapies or other strategies to exploit this weakness in linezolid-resistant MRSA should be pursued.

## MATERIALS AND METHODS

***In vitro* selection experiment.** The MRSA type strain (*Staphylococcus aureus* subsp. *aureus* Rosenbach, ATCC 43300, resistant to oxacillin and methicillin) was used as the ancestral strain for *in vitro* selection. The experiment was modeled on a protocol described by Peleg et al. (31). Briefly, a starting culture in BHI broth was streaked onto BHI agar plates, where a single colony was picked and grown in BHI broth (10 mL) for 24 h with shaking. Then, 20 mL of BHI (containing 0, 2, 4, or 8 $\mu$g/mL linezolid) was inoculated with $10^8$ CFU of MRSA grown under static conditions at 37°C and monitored over several days until the broth became cloudy, indicating bacterial growth. To select bacteria that had acquired a linezolid-resistant phenotype, the culture was diluted in BHI (1:10, 1:100, and 1:1,000) and plated on BHI agar supplemented with either 0, 2, 4, or 8 $\mu$g/mL linezolid. Colonies were counted using a Phenobooth (Singer Instruments, Roadwater, United Kingdom), and the cells were stored frozen at $-80$°C for WGS.

**Bacterial growth and MIC assays.** Growth analysis by monitoring absorbance at 600 nm as a function of time is described by Belousoff et al. (30). MIC assays on the mutants generated in the *in vitro* selection experiments were carried out as previously described using the broth-microdilution method with cation-adjusted Mueller-Hinton (MH) broth as the growth medium (32–34), with linezolid concentrations spanning from 0.0625 to 64 $\mu$g/mL and resistance defined as a MIC of $\geq$8 $\mu$g/mL. Negative growth controls (mock inoculated with sterile MH) and positive growth controls (no antibiotic added) were used to confirm the validity of each MIC. As an internal control for the quality of linezolid, we used the type strain *S. aureus* ATCC 43300 as a reference, because it was used as the ancestor strain in the evolution experiments and we had previously demonstrated that it has an MIC of 1 $\mu$g/mL (11). Results were collated using the Prism software package and the $k_{obs}$ was calculated using the logistic growth curve fit.

**Genome sequencing and analysis.** Eighteen strains (including our starting sample of the ATCC 43300 reference strain) were subjected to genomic DNA (gDNA) sequencing. Next, $\sim$10 ng purified gDNA was prepared with an Illumina Nextera XT kit and sequenced using an Illumina MiSeq PE300. The resulting short-read sequencing libraries were evaluated for quality using FastQC v0.11.7 (35) and quality filters were applied using FastX ToolKit v0.0.13 (36). The genomes were assembled with Unicycler v0.4 (37) and the scaffolding was finalized by RaGOO v1.0.1. SNVs between each pairwise combination were detected using Parsnp (Harvest tool suite v1.1.2) (38). Gingr (Harvest tool suite v1.1.2) (38) was used to generate VCF files, and vcftools was used to summarize the SNV counts. SNPeff v5.0 (39) was used to determinate the effect of the SNV on each gene. Assembly2gene (https://github.com/LPerlaza/Assembly2Gene) was used to identify variants in ribosome genes.

**Purification of 70S ribosomes.** The 70S ribosomes were isolated from MRSA strains ATCC 43300 and N3 as previously described by Belousoff et al. (11).

**Vitrified sample preparation and data collection.** Samples of the 70S ribosomes were concentrated to 0.3 mg/mL. Aliquots of 3 $\mu$L each were applied to a glow-discharged Quantifoil R1.2/1.3 200-mesh holey grid (Quantifoil GmbH, Grosslöbichau, Germany) and flash-frozen in liquid ethane using the Vitrobot Mark IV (Thermo Fisher Scientific), set at 100% humidity and 4°C for the prep chamber with a blot time of 1.5 s. Data were collected on a Glacios transmission electron microscope (TEM; Thermo Fisher Scientific) operating at an accelerating voltage of 200 kV, with a 50-$\mu$m C2 aperture with no objective aperture inserted, and at an indicated magnification of $\times$120 k in nanoprobe TEM mode. A Falcon 3 direct electron detector positioned post-column was used to acquire dose-fractionated images of the samples; the exposure time was 40 s, yielding an electron dose of 47.5 e · Å$^{-2}$. Movies were recorded in gain-normalized mode with the experimental parameters listed in Table 2, using a 9-hole position beam-image shift acquisition pattern in the EPU software package (Thermo Fisher Scientific).

**Data processing.** Micrographs were motion-corrected using UCSF MotionCor2 (40) and dose-weighted averages had their contrast transfer function (CTF) parameters estimated using gCTF (41). Particles were picked using the Laplacian of Gaussian picker in the RELION (v3.1) software package (23, 42, 43). These particles were extracted from the micrographs and subjected to 2D classification, *ab initio* 3D generation, and final 3D refinement in RELION. The resulting homogeneous particle selections then underwent 3D refinement, particle polishing, and CTF envelop fitting in RELION. A mask was then prepared which covered the most of the 50S subunit, and a focused final 3D refinement was performed to yield the final maps of the 50S subunit for each of the two structures.

**Modeling into cryoEM maps.** The coordinates of the 50S subunit from the PDB: 5TCU (11) were used as the starting point for the models described in this work. Manual adjustment of residues which were different from the deposited structure was performed using the coot software package (44) and rigid body-fitting into the cryoEM maps using the SITUS software package (45). The models were then flexibly fitted into the density

using Molecular Dynamics Flexible Fitting (MDFF) as implemented in namd2 (46), and the resulting structures were then further refined in real space using Phenix (47, 48) along with manual curation in coot. See Table 2 for the molecular modeling statistics.

**Model analysis.** Interactions and structure comparisons were carried out using PyMOL Molecular Graphics System v2.3 (Schrödinger, LLC) and figures were generated in either PyMOL or UCSF Chimera (49).

**Data availability.** The structures were deposited and are freely available under the following accession numbers: PDB 7TTU, EMDB EMD-26124 (50S wild type); and PDB 7TTW, EMDB EMD-26125 (50S strain N3).

## SUPPLEMENTAL MATERIAL

Supplemental material is available online only.
**SUPPLEMENTAL FILE 1**, PDF file, 1 MB.
**SUPPLEMENTAL FILE 2**, XLSX file, 0.03 MB.

## ACKNOWLEDGMENTS

This work was supported by grants from US DOD (PRMRP: W81XWH1910126) for M.J.B., the National Health & Medical Research Council of Australia (program grant 1092262 to T.L.), and the Australia Research Council (DP 170103567 to D.W.L.).

We acknowledge support from Thermo Fisher Scientific for the Glacios housed in at the Bio21 Institute, Melbourne University, as part of a collaboration with Patrick Sexton and Denisse Wootten's laboratory. This work was supported by the MASSIVE HPC facility (www.massive.org.au).

The funders had no role in study design, data collection and interpretation, or the decision to submit the work for publication.

L.P.J. and K.-S. T. carried out adaptation experiments and analyzed genomic data. S.J.P., R.M.J., and R.S.B. analyzed cryoEM data and helped with structural modeling. C.J.S. and K.-S. T. performed and analyzed the MIC assays. A.W. and D.L. provided linezolid for selection experiments. T.L. and M.J.B. came up with the experimental design and performed data analysis and interpretation. T.L., M.J.B., R.S.B., and L.P.J. prepared the manuscript. M.J.B. prepared ribosome samples and collected the cryoEM data.

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
