## [Reviewer comments · Microbiology Spectrum]

Microbiology Spectrum

A structurally characterized *Staphylococcus aureus* evolutionary escape route from treatment with the antibiotic linezolid.

Matthew Belousoff, Laura Perlaza-Jimenez, Kher-Shing Tan, Sarah Piper, Rachel Johnson, Rebecca Bamert, Christopher Stubenrauch, Alexander Wright, David Lupton, and Trevor Lithgow

Corresponding Author(s): Matthew Belousoff, Monash University

Review Timeline:

Submission Date:	February 14, 2022
Editorial Decision:	March 8, 2022
Revision Received:	May 18, 2022
Accepted:	June 3, 2022

Editor: Kunyan Zhang

Reviewer(s): The reviewers have opted to remain anonymous.

Transaction Report:

DOI: <https://doi.org/10.1128/spectrum.00583-22>

March 8, 2022

Dr. Matthew J Belousoff
Monash University
Clayton
Australia

Re: Spectrum00583-22 (A structurally characterized *Staphylococcus aureus* evolutionary escape route from treatment with the antibiotic linezolid.)

Dear Dr. Matthew J Belousoff:

Link Not Available

Sincerely,

Kunyan Zhang

Journals Department
Reviewer comments:

Reviewer #1 (Comments for the Author):

This paper describes the characterization by WGS and cryoelectron microscopy of in vitro-selected linezolid (LZ)-resistant first line mutants from one type strain of MRSA. The authors found two mutations in the gene encoding u L3 associated with an increase of the LZ MIC and a modification of the LZ binding site. These results are important to understand the mechanism at the ribosome level to try to prevent the emergence of resistance under antibiotic pressure.

Major remarks

-The results of this study need to be more discussed with the results of the already published studies on staphylococci (both *S.aureus* and Coagulase negative staphylococci) specifically those concerning the change(s) in L3 protein associated with an

increase in LZ MIC and fitness cost.

Line 64: « The time for re-evaluation of this resistance is now". Please discuss and moderate this statement, as linezolid-resistance in MRSA is stable since its introduction in clinical use.

Line 86: please add more recent references on the activity of linezolid on clinical isolates (<1 % of resistance strains)

Line 90: please explain what you mean by "the 14-day timeline".

Line 120. In order to quantify, calculate the generation times of the different mutants from the growth rates in the exponential growth phase. Discuss your results.

Line 193. What do you mean by "genes that permit hypermutation in the *rrl* locus"? Please rephrase

Line 195. Important to discuss this suggestion with the already published studies on LZ-R Staphylococci

Figure 1 B, use hours instead of seconds for the time

Line 147. Please add the 5S rRNA

Line 206. Specify the antibiotic resistance phenotype of the MRSA type strain used in this study.

Line 213: explain what you call a stable linezolid-resistant phenotype.

Line 222: the description of the method for carrying out the MICs being incomplete and imprecise in the reference mentioned, # 27 (Belousoff et al, 2011), please briefly describe the method in the text of the manuscript and specify the strain used as a control. Add the reference of the LZ clinical breakpoint.

Table 2. How do you explain that the same double mutation is associated with both susceptible (LZ MIC=4 mg/l) and resistant (LZ MIC=8 mg/L) phenotypes?

Reviewer #2 (Comments for the Author):

In the manuscript entitled "A structurally characterized *Staphylococcus aureus* evolutionary escape route from treatment with the antibiotic linezolid", Perlaza-Jimenez et al. use a step-wise selection for mutants resistant to linezolid, with an MRSA type strain as the starting point. This is followed by a cryo-EM study to assess the structural basis of resistance to linezolid. Drug-resistant *S. aureus* is a significant problem and worthy of serious study. While this study provides some important insights into this problem, there are some significant shortcomings that should be addressed.

The authors do not make clear why they performed the step-wise selection on increasing drug concentration, rather than selecting for single-step mutants on inhibitory concentrations. A clear explanation for this approach would be helpful. They then use whole genome sequencing to identify all mutations in the resulting strains, and numerous mutations are found throughout the genome. One set of mutations that are common to all the strains (at least that is what this reader infers) are found in ribosomal protein uL3. The authors use cryo-EM to solve the structure of the mutant 50S subunit and uncover the structural changes that are predicted to interfere with linezolid binding. The structural data make sense and are consistent with similar structural studies done with *S. aureus* uL3 mutants resistant to linezolid.

There are several significant issues with the biological work that, if properly addressed, would greatly strengthen the conclusions of this study. One thing that is very confusing is the fact that, according to Table 2, lineages with the G155R mutation alone (A1 through C1) are designated as susceptible. The only resistant lineages (K3 through R3) are those with the G155R/M169I double mutation. However, a number of lineages (D2 through J2) with this same double mutation are designated as susceptible. These data would certainly seem to indicate that the double mutation they have identified is not sufficient for the resistance phenotype. All of this is complicated by the numerous mutations elsewhere in the genome. The authors state that "in order to establish themselves for growth in the presence of linezolid, the three first generation strains have 45 SNVs in common." It would thus appear that the relationship between the uL3 mutations and resistance cannot be derived from the current data. This also applies to interpretation of the growth rate data. The authors perform doubling time measurements of A1 (susceptible uL3-G155R) and N3 (resistant uL3-G155R/M169I) to show a growth defect caused by the uL3 mutations, but do not obtain a doubling time for any of the susceptible uL3-G155R/M169I. Do these also have a growth defect? If not, how can they make any conclusions about the nature of the growth defect? Any one of the numerous SNVs could be causing the slower growth. It would seem to be important to perform more extensive growth rate measurements on a number of the strains (both susceptible and resistant) before making any conclusions. If the uL3 mutations are responsible, then all the double mutants should have this same defect.

The structural work does show convincing changes that would explain the resistance phenotype. It is also consistent with previous structures of similar mutations. However, the weaknesses in the phenotypic work outlined above are problematic.

Minor comments

1. It would help the reader if the authors were more careful about how they refer to their strains. For instance, strain 'A1/B1/C1' as written implies that it is a single strain, not three independent strains as indicated in Tables 1 and 2. It is also unclear what this designation is meant to indicate and how it relates to Fig 1. Where did each strain come from? What cycle of selection? This needs to be stated more explicitly.
2. The authors use the term "generation", which is confusing as this term is generally used to refer to a doubling of a bacterial culture. Perhaps "cycle" or "selection cycle" would be more appropriate.
3. In any discussion of evolution of fitness-enhancing mutations, the work of Dan Andersson should be cited.
4. Table 1 and Table 2 could be combined into a single Table.

5. The antibiotic "oxazolidinone" is misspelled as "oxazolidone" throughout the manuscript.
6. In numbering 23S rRNA nucleotides, the authors should use 'U' instead of 'T', as in 'G2576T' should read 'G2576U' (line 82).
7. The authors state (line 147/148) that "The 70S ribosome in *S. aureus* is composed of 46 ribosomal proteins and the 16S rRNA and 23S rRNA." Please ammend to include 5S rRNA.
8. Typo (line 187): "lead" should read "led".
9. Typo (line195): "determinate" should read "determinant".
10. The first sentence in the Materials and Methods (line 226) does not make sense. Looks like a spontaneous deletion.

Reviewer #3 (Public repository details (Required)):

The structures and maps should be deposited at the PDB

Reviewer #3 (Comments for the Author):

The paper presents a predicted antibiotic resistant structure and confirms the speculation on how the resistance achieved. Although this is not a revolutionary paper, this still has a medicinal impact.

My comments:

- There is a need of showing that the uL3 mutation is the reason for the linezolid resistance. It is possible that the resistance is gained through another cellular mechanism, for example- a defect in transporting the linezolid inside the cell or an enzyme which modify the linezolid somehow. It is not convincing that the effect of the resistance caused from the ribosome different structure.
- Why did the authors decided not to determined the structure of the uL3 with only G155R mutation? the 169 mutation is quit far from the PTC...
- uL3 mutations cause also resistance to other ribosomal antibiotics families. This should be written in the text (with the correct references) and explained according to the author' findings.
- "Ribosomal protein uL3 appears to be a key element for imparting linezolid resistance"- this statement is not correct. In the literature the are evidences for mutations in ribosomal protein uL4 and rRNA as well.
- There is not fig comparing between nucleotide 2512 before and after the mutation.
- Figure 1A- below GEN3, 3 lower plates, there is another antibiotic concentration written there. It is not clear what is it.
- Please add to the text few words explaining what is the meaning of resistance breakpoint.
- Figure 1B- why only A1 and N3 growth curved are shown? What about the other strains and how do they relate to the scheme in figure 1A? This information belongs to the supplementary data and not to the main text.
- Figure 3A- Ribosomal proteins and rRNA are presented in very similar colors, it is hard to distinguish between.
- Figure 3B- labels for the residues are missing. The linezolid is not actually there, only for marking the binding pocket. This should be stated or presented differently.
- Figure 3B- a 2D rRNA map of the PTC with helices and residues numbers is missing.
- Figure 3C+D- should be merged as an overlay of the 2 structure, in order to emphasize the differences between the structures.
- Figure 3D- typo: "stain" instead of "strain".
- Figure 3E- why showing tedizolid and not linezolid? Also here the drug in not really part of the structure.
- Figure 3- an electron density maps of the 2 structures and the different residues are missing. If the electron density in these area is not clear, this should be mentioned and explained as possible flexibility of this residue.
- Figure 3E- a distance measurement is missing.
- A cryo-EM flow chart with micrographs and classes is missing.
- Table 3- RNA backbone validation statistics is missing.

Staff Comments:

Preparing Revision Guidelines

- Point-by-point responses to the issues raised by the reviewers in a file named "Response to Reviewers," NOT IN YOUR COVER LETTER.

- Upload a compare copy of the manuscript (without figures) as a "Marked-Up Manuscript" file.
- Each figure must be uploaded as a separate file, and any multipanel figures must be assembled into one file.
- Manuscript: A .DOC version of the revised manuscript
- Figures: Editable, high-resolution, individual figure files are required at revision, TIFF or EPS files are preferred

Please return the manuscript within 60 days; if you cannot complete the modification within this time period, please contact me. If you do not wish to modify the manuscript and prefer to submit it to another journal, please notify me of your decision immediately so that the manuscript may be formally withdrawn from consideration by Microbiology Spectrum.

In the manuscript entitled "A structurally characterized *Staphylococcus aureus* evolutionary escape route from treatment with the antibiotic linezolid", Perlaza-Jimenez et al. use a step-wise selection for mutants resistant to linezolid, with an MRSA type strain as the starting point. This is followed by a cryo-EM study to assess the structural basis of resistance to linezolid. Drug-resistant *S. aureus* is a significant problem and worthy of serious study. While this study provides some important insights into this problem, there are some significant shortcomings that should be addressed.

The authors do not make clear why they performed the step-wise selection on increasing drug concentration, rather than selecting for single-step mutants on inhibitory concentrations. A clear explanation for this approach would be helpful. They then use whole genome sequencing to identify all mutations in the resulting stains, and numerous mutations are found throughout the genome. One set of mutations that are common to all the strains (at least that is what this reader infers) are found in ribosomal protein uL3. The authors use cryo-EM to solve the structure of the mutant 50S subunit and uncover the structural changes that are predicted to interfere with linezolid binding. The structural data make sense and are consistent with similar structural studies done with *S. aureus* uL3 mutants resistant to linezolid.

There are several significant issues with the biological work that, if properly addressed, would greatly strengthen the conclusions of this study. One thing that is very confusing is the fact that, according to Table 2, lineages with the G155R mutation alone (A1 through C1) are designated as susceptible. The only resistant lineages (K3 through R3) are those with the G155R/M169I double mutation. However, a number of lineages (D2 through J2) with this same double mutation are designated as susceptible. These data would certainly seem to indicate that the double mutation they have identified is not sufficient for the resistance phenotype. All of this is complicated by the numerous mutations elsewhere in the genome. The authors state that "in order to establish themselves for growth in the presence of linezolid, the three first generation strains have 45 SNVs in common." It would thus appear that the relationship between the uL3 mutations and resistance cannot be derived from the current data. This also applies to interpretation of the growth rate data. The authors perform doubling time measurements of A1 (susceptible uL3-G155R) and N3 (resistant uL3-G155R/M169I) to show a growth defect caused by the uL3 mutations, but do not obtain a doubling time for any of the susceptible uL3-G155R/M169I. Do these also have a growth defect? If not, how can they make any conclusions about the nature of the growth defect? Any one of the numerous SNVs could be causing the slower growth. It would seem to be important to perform more extensive growth rate measurements on a number of the strains (both susceptible and resistant) before making any conclusions. If the uL3 mutations are responsible, then all the double mutants should have this same defect.

The structural work does show convincing changes that would explain the resistance phenotype. It is also consistent with previous structures of similar mutations. However, the weaknesses in the phenotypic work outlined above are problematic.

Minor comments

1. It would help the reader if the authors were more careful about how they refer to their strains. For instance, strain 'A1/B1/C1' as written implies that it is a single strain, not three independent strains as indicated in Tables 1 and 2. It is also unclear what this designation is meant to indicate and how it relates to Fig 1. Where did each strain come from? What cycle of selection? This needs to be stated more explicitly.
2. The authors use the term "generation", which is confusing as this term is generally used to refer to a doubling of a bacterial culture. Perhaps "cycle" or "selection cycle" would be more appropriate.
3. In any discussion of evolution of fitness-enhancing mutations, the work of Dan Andersson should be cited.
4. Table 1 and Table 2 could be combined into a single Table.
5. The antibiotic "oxazolidinone" is misspelled as "oxazolidone" throughout the manuscript.
6. In numbering 23S rRNA nucleotides, the authors should use 'U' instead of 'T', as in 'G2576T' should read 'G2576U' (line 82).
7. The authors state (line 147/148) that "The 70S ribosome in *S. aureus* is composed of 46 ribosomal proteins and the 16S rRNA and 23S rRNA." Please amend to include 5S rRNA.
8. Typo (line 187): "lead" should read "led".
9. Typo (line 195): "determinate" should read "determinant".
10. The first sentence in the Materials and Methods (line 226) does not make sense. Looks like a spontaneous deletion.

RESPONSE TO REVIEWERS

Reviewer #1 (Comments for the Author):

This paper describes the characterization by WGS and cryoelectron microscopy of in vitro-selected linezolid (LZ)-resistant first line mutants from one type strain of MRSA. The authors found two mutations in the gene encoding uL3 associated with an increase of the LZ MIC and a modification of the LZ binding site. These results are important to understand the mechanism at the ribosome level to try to prevent the emergence of resistance under antibiotic pressure.

Major remarks

-The results of this study need to be more discussed with the results of the already published studies on staphylococci (both S.aureus and Coagulase negative staphylococci) specifically those concerning the change(s) in L3 protein associated with an increase in LZ MIC and fitness cost.

Several comments by the reviewers called for more discussion of our results and, in response, we have separated out a distinct Discussion section in the revised manuscript.

There are very few papers dealing with uL3 mutations in Staphylococcus, and the phenotypic consequences for various different mutations vary from little growth defect (10.1089/mdr.2016.0137) to very large growth defects for a G144D mutation (10.1128/AAC.00179-15). We have included these various studies in the expanded Discussion (p. 8, lines 10-29).

Line 64: « The time for re-evaluation of this resistance is now". Please discuss and moderate this statement, as linezolid-resistance in MRSA is stable since its introduction in clinical use.

We have adjusted the text (p. 3, lines 15-16), the line:

“Given that linezolid- and tedizolid-resistance is considered to be uncommon relative to other AMR phenotypes in MRSA^{5,7}, the time for re-evaluation of this resistance is now” now reads:

“Linezolid- and tedizolid-resistance is considered to be uncommon relative to other AMR phenotypes in MRSA^{5,7}, providing limited opportunities to understand any constraints and the relative fitness of strains resistant to oxazolidinone antibiotics”

Line 86: please add more recent references on the activity of linezolid on clinical isolates (<1 % of resistance strains)

A recent meta analysis of clinical data has been cited in the text now (p. 4, lines 8-9), in support of the observation that linezolid-resistant strains are not common.

Line 90: please explain what you mean by" the 14-day timeline".

We have modified the text (p. 4, line 15) to remove this statement of the duration of the experiment.

Line 120. In order to quantify, calculate the generation times of the different mutants from the growth rates in the exponential growth phase. Discuss your results.

We have calculated a k_{obs} (growth rate) value for each strain. An additional Supplemental Table S2 has been added, reporting the observed growth rate constants for each bacterial strain. We have also added text into the MATERIALS AND METHODS section (p. 9-10, lines 32- 4) as well as performed growth phenotype assays on more of the strains to better represent the growth phenotypes of the strains.

Line 193. What do you mean by" genes that permit hypermutation in the rrl locus"? Please rephrase

We had referred to a previous study that had identified a *recQ* missense mutation (Glu69Asp) associated with hypermutation that may have facilitated recombination among the multiple *rrn* loci (doi:10.1371/journal.pone.0155512). In revision of the text, reference to this work has been deleted.

Line 195. Important to discuss this suggestion with the already published studies on LZ-R Staphylococci

We have expanded the discussion of this point in the revised text, and added a key finding by Locke and coworkers which reports some uL3 mutations in combination with a more potent *cfr* gene (p. 8, lines 31-33). We discuss the possibility that the uL3 mutations come first, bridging the gap before the acquisition of a more potent resistance mechanism mediated through controlled methylation such as is enacted via *cfr*.

Figure 1 B, use hours instead of seconds for the time

We have adjusted the plots with the X-axis in hours instead of seconds.

Line 147. Please add the 5S rRNA

This has been corrected (p. 6, lines 32).

Line 206. Specify the antibiotic resistance phenotype of the MRSA type strain used in this study.

It has been added (p. 9, lines 19-20).

Line 213: explain what you call a stable linezolid-resistant phenotype.

We have deleted the word “stable” for clarity.

Line 222: the description of the method for carrying out the MICs being incomplete and imprecise in the reference mentioned, # 27 (Belousoff et al, 2011), please briefly describe the method in the text of the manuscript and specify the strain used as a control. Add the reference of the LZ clinical breakpoint.

We have adjusted the text. The reference strain and other details are now noted in the METHODS section (p. 10, lines 2-3).

Table 2. How do you explain that the same double mutation is associated with both susceptible (LZ MIC=4 mg/l) and resistant (LZ MIC=8 mg/L) phenotypes?

The MIC assessment is done across two-fold dilutions of drug, so the Generation 2 (4 µg/mL) and Generation 3 (8 µg/mL) straddle the break-point: i.e. Generation 2 is almost resistant, Generation 3 is just barely resistant, in terms of the clinical breakpoint value. We discuss one potential explanation for how this might be enacted epigenetically, given that there is no genetic explanation for the final step to resistance on 8 µg/mL linezolid (p. 8-9). We have reviewed and reinspected our genomics data and there is no genetic explanation as to why this is the case. The epigenetic explanation is also consistent with the dissimilar aggregation to the growth rate seen in the various Generation 3 strains, as measured for some in Figure 1B.

Reviewer #2 (Comments for the Author):

In the manuscript entitled "A structurally characterized Staphylococcus aureus evolutionary escape route from treatment with the antibiotic linezolid", Perlaza-Jimenez et al. use a step-wise selection for mutants resistant to linezolid, with an MRSA type strain as the starting point. This is followed by a cryo-EM study to assess the structural basis of resistance to linezolid. Drug-resistant S. aureus is a significant problem and worthy of serious study. While this study provides some important insights into this problem, there are some significant shortcomings that should be addressed.

The authors do not make clear why they performed the step-wise selection on increasing drug

concentration, rather than selecting for single-step mutants on inhibitory concentrations. A clear explanation for this approach would be helpful.

We have used the new DISCUSSION section to justify our approach (p. 7, lines 29-33). We appreciate the reviewer's suggestion here. We had followed a similar approach to Peleg and co-workers (doi:10.1371/journal.pone.0028316 (2012) that worked for investigating daptomycin resistance. However, when we attempted a single-step selection with a high initial concentration of linezolid, we never observed any break through growth in statically grown cultures, and never obtained colonies when the culture medium was plated out onto Agar-BHI-linezolid plates. We therefore decided to perform the experiment in a step wise fashion, allowing the *S. aureus* to evolve under less selection pressure.

*They then use whole genome sequencing to identify all mutations in the resulting stains, and numerous mutations are found throughout the genome. One set of mutations that are common to all the strains (at least that is what this reader infers) are found in ribosomal protein uL3. The authors use cryo-EM to solve the structure of the mutant 50S subunit and uncover the structural changes that are predicted to interfere with linezolid binding. The structural data make sense and are consistent with similar structural studies done with *S. aureus* uL3 mutants resistant to linezolid.*

There are several significant issues with the biological work that, if properly addressed, would greatly strengthen the conclusions of this study. One thing that is very confusing is the fact that, according to Table 2, lineages with the G155R mutation alone (A1 through C1) are designated as susceptible. The only resistant lineages (K3 through R3) are those with the G155R/M169I double mutation. However, a number of lineages (D2 through J2) with this same double mutation are designated as susceptible. These data would certainly seem to indicate that the double mutation they have identified is not sufficient for the resistance phenotype. All of this is complicated by the numerous mutations elsewhere in the genome. The authors state that "in order to establish themselves for growth in the presence of linezolid, the three first generation strains have 45 SNVs in common." It would thus appear that the relationship between the uL3 mutations and resistance cannot be derived from the current data.

We appreciate the point. Since the MIC assessment is done across two-fold dilutions of drug, so the Generation 2 (4 µg/mL) and Generation 3 (8 µg/mL) straddle the break-point: i.e. Generation 2 is almost resistant, Generation 3 is just barely resistant, in terms of the clinical breakpoint value. We now explicitly draw attention to why Strain N3 was chosen for assessment of its ribosomes (p. 6, lines 34), highlighting the importance of the ribosomal protein uL3 mutations.

We discuss one potential explanation for how this might be enacted epigenetically (p. 8, lines 27-29), given that there is no genetic explanation for the final step to resistance on 8 µg/mL linezolid. We have reviewed and reinspected our genomics data and there is no genetic explanation as to why this is the case. The epigenetic explanation is also consistent with the dissimilar aggregation to the growth rate seen in the various Generation 3 strains measured in Figure 1B.

This also applies to interpretation of the growth rate data. The authors perform doubling time measurements of A1 (susceptible uL3-G155R) and N3 (resistant uL3-G155R/M169I) to show a growth defect caused by the uL3 mutations, but do not obtain a doubling time for any of the susceptible uL3-G155R/M169I. Do these also have a growth defect? If not, how can they make any conclusions about the nature of the growth defect? Any one of the numerous SNVs could be causing the slower growth. It would seem to be important to perform more extensive growth rate measurements on a number of the strains (both susceptible and resistant) before making any conclusions. If the uL3 mutations are responsible, then all the double mutants should have this same defect.

We appreciate this point and have performed more growth phenotype assays. The primary data is now shown in Figure 1B and the quantification of the growth rates shown in new Table S1.

The revised text (p. 5, lines 18-31) is a clearer link to the observations made.

The structural work does show convincing changes that would explain the resistance phenotype. It is

also consistent with previous structures of similar mutations. However, the weaknesses in the phenotypic work outlined above are problematic.

We appreciate this and have tried to make a clearer link between the genotype complexities in the evolution pathway and clear structure-informed phenotype.

Minor comments

1. It would help the reader if the authors were more careful about how they refer to their strains. For instance, strain 'A1/B1/C1' as written implies that it is a single strain, not three independent strains as indicated in Tables 1 and 2. It is also unclear what this designation is meant to indicate and how it relates to Fig 1. Where did each strain come from? What cycle of selection? This needs to be stated more explicitly.

We have revised the text accordingly and added a Supplemental Figure S1 to more clearly represent the strain naming convention used in the manuscript to address these points. We also modified the labels in Figure 1 and 2 to be consistent with the strain naming convention.

2. The authors use the term "generation", which is confusing as this term is generally used to refer to a doubling of a bacterial culture. Perhaps "cycle" or "selection cycle" would be more appropriate.

We have now been explicit in the use of the terminology for Generation 1 strains, Generation 2 strains and Generation 3 strains. Used in conjunction with the numerals there would not be any ambiguity in the terminology.

3. In any discussion of evolution of fitness-enhancing mutations, the work of Dan Andersson should be cited.

Our study is not focussed on fitness-enhancing mutations. We have read through the works of Dan Andersson but were unable to find a relevant paper to cite. We are open to being informed of important papers that we may have missed in the literature review.

4. Table 1 and Table 2 could be combined into a single Table.

This has been done.

5. The antibiotic "oxazolidinone" is misspelled as "oxazolidone" throughout the manuscript.

This has been corrected.

6. In numbering 23S rRNA nucleotides, the authors should use 'U' instead of 'T', as in 'G2576T' should read 'G2576U' (line 82).

This has been fixed.

7. The authors state (line 147/148) that "The 70S ribosome in S. aureus is composed of 46 ribosomal proteins and the 16S rRNA and 23S rRNA." Please amend to include 5S rRNA.

This has been corrected.

8. Typo (line 187): "lead" should read "led".

Corrected.

9. Typo (line 195): "determinate" should read "determinant".

Corrected

10. The first sentence in the Materials and Methods (line 226) does not make sense. Looks like a spontaneous deletion.

This has been corrected.

Reviewer #3 (Public repository details (Required)):

The structures and maps should be deposited at the PDB

These structures have already been deposited (The structures were deposited and are freely available with accession codes, PDB: 7TTU, EMDB: EMD-26124 (50S-WT) and PDB: 7TTW, EMDB: EMD-26125 (50S-StrainN3)). Upon the publication of the paper, the coordinates and maps will be released.

Reviewer #3 (Comments for the Author):

The paper presents a predicted antibiotic resistant structure and confirms the speculation on how the resistance achieved. Although this is not a revolutionary paper, this still has a medicinal impact.

My comments:

- There is a need of showing that the uL3 mutation is the reason for the linezolid resistance. It is possible that the resistance is gained through another cellular mechanism, for example- a defect in transporting the linezolid inside the cell or an enzyme which modify the linezolid somehow. It is not convincing that the effect of the resistance caused from the ribosome different structure.

- Why did the authors decided not to determined the structure of the uL3 with only G155R mutation? the 169 mutation is quit far from the PTC...

This is a fair point and we did originally attempt a structure for the strain with the G155R single mutation. Unfortunately, the quality of the maps in the structure were not of publication quality. We are satisfied that the comparison of the starting strain and the double mutation in uL3 were sufficient to explain the rearrangement of the linezolid binding site.

- uL3 mutations cause also resistance to other ribosomal antibiotics families. This should be written in the text (with the correct references) and explained according to the author' findings.

We agree that the uL3 also has the potential to cause cross resistance to other antibiotics that bind in the PTC. We have also added to the text that the G155R mutation has been reported to confer resistance to Tiamulin. We have added these comments to the manuscript (p. 6, lines 29-30).

- "Ribosomal protein uL3 appears to be a key element for imparting linezolid resistance"- this statement is not correct. In the literature there are evidences for mutations in ribosomal protein uL4 and rRNA as well.

We agree and have added this to the text (p. 3, lines 32-34).

- There is not fig comparing between nucleotide 2512 before and after the mutation.

This is presented in Figure 3C and 3D, showing the change in position of C2512. This figure panels are from the exact same viewing position. See explanation below.

- Figure 1A- below GEN3, 3 lower plates, there is another antibiotic concentration written there. It is not clear what is it.

We have removed the extra antibiotic concentrations from the figure as we agree that it was confusing. They initially were there to highlight which static growth cultures were plated onto the selection plates.

- Please add to the text few words explaining what is the meaning of resistance breakpoint.

We have added a sentence in the text (p. 6, lines 30-32) explaining the definition.

- Figure 1B- why only A1 and N3 growth curved are shown? What about the other strains and how do they relate to the scheme in figure 1A? This information belongs to the supplementary data and not to the main text.

We appreciate this point, which was also raised by Reviewer 1. We have added more data to Figure 1B, to show growth rate data for strains from all three generations, including each step that led to Strain N3 (the strain for which the ribosome structure was determined).

- Figure 3A- Ribosomal proteins and rRNA are presented in very similar colors, it is hard to distinguish between.

We do appreciate this point. Unfortunately, the complexity of the ribosome structure means that using radically different colors makes the figure really distracting. Our Figure 3A is simply an overview of where in the large subunit the drug binds. We did not aim to clearly show the difference between the proteins and rRNA in this representation.

-Figure 3B- labels for the residues are missing. The linezolid is not actually there, only for marking the binding pocket. This should be stated or presented differently.

We have not labelled everything for the sake of clarity. We attempted a few means for labelling all the residues, but it makes the figure harder to interpret. We do appreciate the need to make clear that this is a modelled drug position for graphical needs and we have modified the text in the Figure legend accordingly.

- Figure 3B- a 2D rRNA map of the PTC with helices and residues numbers is missing.

We have added another figure Panel to Figure 3. Showing the Domain V part of the 23S rRNA, focussed on the peptidyl transferase centre, to make it clearer to the reader what parts of the ribosome we are referring to.

- Figure 3C+D- should be merged as an overlay of the 2 structure, in order to emphasize the differences between the structures.

We appreciate the suggestion, and we have tried to overlay. However, the overlay of these two complex figures is overly busy and hard to interpret. Instead, we now make clear (p. 16, lines 37-38) that Figures 3C and 3D are viewed from the exact same viewing angle, so that they are completely comparable.

-Figure 3D- typo: "stain" instead of "strain".

Corrected

- Figure 3E- why showing tedizolid and not linezolid? Also here the drug is not really part of the structure.

In past studies we have been criticized for focussing on linezolid without showing the how the newest oxazolidinone drugs bind in a similar way. Since we are simply overlaying the position of the drugs to orient the reader as to the area of interest in the ribosome, we hope that one example with linezolid and one with tedizolid will quell critical comments.

We have revised the Figure legend to indicate more clearly that the antibiotic is not present in the structures, but that it is a model to help orient the reader.

- Figure 3- an electron density maps of the 2 structures and the different residues are missing. If the electron density in these area is not clear, this should be mentioned and explained as possible flexibility of this residue.

We appreciate this point in principle, but the density around this region is unambiguous. The data is available for experts to evaluate (Accession codes, PDB: 7TTU, EMDB: EMD-26124 (50S-WT) and PDB: 7TTW, EMDB: EMD-26125 (50S-StrainN3).)

- Figure 3E- a distance measurement is missing.

We made no discussion of atomic distances, so they are not presented in the Figure.

- A cryo-EM flow chart with micrographs and classes is missing.

CryoEM data processing is routine now and we have not included the generic work flow in publications for several years now. If this is needed, a chart with micrographs can be provided as we have done some years ago.

-Table 3- RNA backbone validation statistics is missing.

We have added this data in an additional row to the new Table 2 (was previously Table 3). We appreciate the importance of this addition.

June 3, 2022

Dr. Matthew J Belousoff
Monash University
Clayton
Australia

Re: Spectrum00583-22R1 (A structurally characterized *Staphylococcus aureus* evolutionary escape route from treatment with the antibiotic linezolid.)

Dear Dr. Matthew J Belousoff:

Your manuscript has been accepted, and I am forwarding it to the ASM Journals Department for publication. You will be notified when your proofs are ready to be viewed.

Sincerely,

Kunyan Zhang
Editor, Microbiology Spectrum

Journals Department
Supplemental Table 2: Accept
Supplemental Material: Accept

Re-review Spectrum00583-22

Comments for Authors

The revised manuscript is much improved and addresses most issues. My main concern was the large number of SNVs in all the derivatives combined with the variation in growth and resistance phenotypes of all the strains, making it difficult to establish the role of the uL3 mutations in these properties. The authors have toned down this aspect of their conclusions, leaving the main conclusions to derive from the structural data. This conclusion is reasonable. Ideally the authors should have performed allelic exchange to make strains with only the uL3 mutations and this remains a major weakness of the study. Nevertheless, the structural data themselves are significant.